Introspective analysis of convolutional neural networks for improving discrimination performance and feature visualisation

Shafiq Shakeel
Azim Tayyaba tayyaba.azim@imsciences.edu.pk
Center of Excellence in IT, Institute of Management Sciences (IMSciences) , Peshawar, KPK , Pakistan
Asif Muhammad
Electronic publication date: 2021 May 4
Publication date: 2021
Volume: 7
Electronic Location ID: e497
Received 2020 Oct 28; Accepted 2021 Mar 30
Copyright: © 2021 Shafiq and Azim
Copyright year: 2021
Copyright holder: Shafiq and Azim
License: This is an open access article distributed under the terms of the Creative Commons Attribution License, which permits unrestricted use, distribution, reproduction and adaptation in any medium and for any purpose provided that it is properly attributed. For attribution, the original author(s), title, publication source (PeerJ Computer Science) and either DOI or URL of the article must be cited.
License URL: https://creativecommons.org/licenses/by/4.0/

Keywords: Local binary pattern (LBP), Support vector machines (SVM), k-nearest neighbour (k-NN), Convolutional neural network (CNN), Discriminatively boosted alternative to pooling (DBAP) layer, Improved LeNet, Improved AlexNet, Feature visualistion, Tensorflow, Multi-class classification

Funding: The authors received no funding for this work.

==============================
Deep neural networks have been widely explored and utilised as a useful tool for feature extraction in computer vision and machine learning. It is often observed that the last fully connected (FC) layers of convolutional neural network possess higher discrimination power as compared to the convolutional and maxpooling layers whose goal is to preserve local and low-level information of the input image and down sample it to avoid overfitting. Inspired from the functionality of local binary pattern (LBP) operator, this paper proposes to induce discrimination into the mid layers of convolutional neural network by introducing a discriminatively boosted alternative to pooling (DBAP) layer that has shown to serve as a favourable replacement of early maxpooling layer in a convolutional neural network (CNN). A thorough research of the related works show that the proposed change in the neural architecture is novel and has not been proposed before to bring enhanced discrimination and feature visualisation power achieved from the mid layer features. The empirical results reveal that the introduction of DBAP layer in popular neural architectures such as AlexNet and LeNet produces competitive classification results in comparison to their baseline models as well as other ultra-deep models on several benchmark data sets. In addition, better visualisation of intermediate features can allow one to seek understanding and interpretation of black box behaviour of convolutional neural networks, used widely by the research community.

Introduction

Deep learning architectures such as convolutional neural networks, recurrent neural networks and deep belief networks have been applied to a wide range of applications in domains such as natural language processing, speech recognition, computer vision, and bioinformatics, where they have produced outstanding results approximately the same and in some scenarios better than the humans (He et al., 2015; Silver et al., 2016; LeCun et al., 1990; Szegedy et al., 2015; Girshick et al., 2014; Hinton et al., 2012; Yu, Liu & Mao, 2018; Zhang et al., 2016; Masumoto et al., 2019; Le & Nguyen, 2019; Le, 2019; Do, Le & Le, 2020). Among these deep models, convolutional neural network (CNN) is the most popular choice for automatically learning visually discriminative features memorised by the fully connected layers. The interest of researchers in CNN triggered when Krizhevsky, Sutskever & Hinton (2012) showed record beating performance on ImageNet 2012 object classification data set with their CNN (AlexNet), achieving an error rate of 16.4% in comparison to 26.1% error shown by the runner up. Ever since then, various variants of deep convolutional models such as Visual Geometry Group (VGG)-Very Deep (VD) model (Simonyan & Zisserman, 2014), GoogLeNet/Inception (Szegedy et al., 2015) and ResNet (He et al., 2016) have been introduced, increasing the depth of the models from eight layers in AlexNet to 152 layers in ResNet. These models have not just progressed in depth but also their intricacy of connectivity, type of activation function and the training algorithm that prevents the diminishing gradient issue observed during training through back propagation in ultra deep models.

Keeping in account the success of deep neural models, many researchers have treated CNN as a black box feature extractor where end-to-end learning framework is utilised to draw discriminative features from the last fully connected (FC) layers. The last fully connected layers are successfully utilised to extract global image descriptors as they possess rich high level semantic information that can effectively distinguish the object of interest from the background (Sharif Razavian et al., 2014). In contrast, the intermediate layers of CNN are popular for extracting spatial and local characteristics of images which are important to extract expressive regions of objects, yet cannot serve very well as a global image descriptor. To get improved performance on different classification tasks, most of the researchers have focused on increasing the depth of the convolutional neural model and varied the network’s training strategy and activation functions (Bengio, Courville & Vincent, 2013; LeCun, Bengio & Hinton, 2015). We observe that designing and training such an ultra-deep architecture for global feature extraction is: (i) Expensive in terms of computation, (ii) results in model size large in terms of disk space utilisation and memory usage, (iii) prone to overfitting when the data set size is limited, and (iv) requires a large amount of labelled training data when fine tuning the model for new application domains. On account of these challenges, we take an introspective approach to understand the functionality and behaviour of the intermediate layers, in particular the convolutional and pooling layers of CNN, and propose a novel technique to improve their representational power along with increasing the discrimination performance of the model without going deeper with additional hidden layers. Visualisation of features allows one to functionally understand and interpret the behaviour of deep model’s internal layers in connection to its progressive depth and output (Raghu et al., 2017; Poole et al., 2016). Visualising the representational capacity of deep models has been a topic of recent interest to express general priors about the world that can help one identify the stimuli causing a certain output and ultimately design learning machines that can solve different AI tasks (Bengio, Courville & Vincent, 2013; Zeiler & Fergus, 2013; Mordvintsev, Olah & Tyka, 2015; Raghu et al., 2017; Santos & Abel, 2019). The topic has grabbed interest of the research community so much so that dedicated workshops in leading conferences like NIPS and CVPR are arranged to discuss the works under this theme. We have therefore laid our focus on developing a technique that can transform features from model’s intermediate layers into a visually powerful tool for introspective analysis, as well as act as discriminative off the shelf feature extractor for image classification with simple and sophisticated machine learning classifiers. Our empirical results reveal that with the proposed technique, intermediate layers close to the input layer could also be made more competent for feature visualisation and discrimination tasks.

The main contributions of this work are outlined as follows: (1) Improving the classification performance of classical CNN architectures: LeNet and AlexNet on benchmark data sets without increasing their depth (hidden layers), (2) Improving the visualisation power of features learned by the intermediate layers of CNN, (3) Introducing discriminatively boosted alternative to pooling (DBAP) layer in the CNN architectures, that can serve independently as an efficient feature extractor for classification when used with classifiers such as k-nearest neigbour (k-NN) and support vector machines (SVM). The pretrained CNN with DBAP layer offers features that could be deployed in resource constrained environments where ultra-deep models could not be stored, retrieved and trained.

The remaining paper is structured as follows: “Related Work” discusses the related research work carried out in the area of computer vision. “Preliminaries” provides preliminary information required to understand the details of proposed methodology discussed in “Methodology”. “Experiments and Results” discusses the benchmark data sets, implementation details and evaluates the results of conducted experiments. We conclude this work in “Conclusion & Future Work” with a discussion on the future work intended to further improve and extend this research in future. There is also a Appendix section (Appendix) that holds additional results to provide in depth analysis of the proposed change in convolutional neural models.

Related work

There has been a recent surge of interest in understanding and visualising the intermediate layers of deep models for interpretability and explainability, leading to the development of more stable and reliable machine learning systems (Zeiler & Fergus, 2014; Ren et al., 2019; Bau et al., 2019; Hazard et al., 2019; Gagne et al., 2019; Hohman et al., 2018). The visualisation techniques allow the researchers and practitioners understand what features are being learned by the deep model at each stage. Visualisation diagnostics may also serve as an important debugging tool to improve a model’s performance, make comparisons and select optimal model parameters for the task at hand. This often requires monitoring the model during the training phase, identifying misclassified examples and then testing the model on a handful of well-known data instances to observe performance. Generally, the following parameters of deep model are visualised either during or after the training phase: (1) Weights on the neural connections (Smilkov et al., 2017), (2) convolutional filters (Zeiler & Fergus, 2014; Yosinski et al., 2015) (3) neuron activations in response to a single or group of instances (Goodfellow, Bengio & Courville, 2016; Yosinski et al., 2015), (4) gradients for the measurement and distribution of train error (Cashman et al., 2017), and (5) model metrics such as loss and accuracy computed at each epoch. This work focuses on improving the visualisation power of deep neural models in addition to enhancing their discrimination ability as a classifier and feature extractor.

The fully connected (FC) layers of deep convolutional neural network have often been utilised to extract features due to their higher discriminative ability and semantic representation of image concepts that makes them a powerful global descriptor (Simonyan & Zisserman, 2014; He et al., 2016). The FC features have demonstrated their advantage over Vector of Locally Aggregated Descriptors (VLAD) and Fisher vector descriptors and are known to be invariant to illumination and rotation to some extent, however they lack the description of local patterns captured by the convolutional layers. To address this limitation, some researchers have proposed to utilise the intermediate layers of deep models to improve their performance on various tasks (Cimpoi, Maji & Vedaldi, 2015; Babenko & Lempitsky, 2015; Liu et al., 2017; Yue-Hei Ng, Yang & Davis, 2015; Liu, Shen & Van Hengel, 2015). For instance, Ng, Yang & Davis (2015) aggregated convolutional layer activations using vector of locally aggregated descriptors (VLAD) and achieved competitive performance on image retrieval task. Tolias, Sicre & Jégou (2015) max pooled the activations of the last convolutional layer to represent each image patch and achieved compelling performance for object retrieval. Liu, Shen & Van Hengel (2017) built a powerful image representation using activations from two consecutive convolutional layers to recognise images. Kumar, Banerjee & Vemuri (2009) and Kumar et al. (2012) introduced the use of Volterra theory for the first time to learn discriminative convolution filters (DCF) from the pixel features on gray-level images.

In addition to the convolutional layers, researchers have also explored the use of various types of pooling functions from simple ones such as max, average, and stochastic pooling to complex ones, like spatial pyramid pooling network (SPP-Net), which allows the convolutional neural model to take images of variable scales using spatial pyramid aggregation scheme (He et al., 2014). The pooling layers have traditionally been utilised in CNN to avoid overfitting by reducing the size of the detected features by a factor of two. However, the fact that they lose spatial information and keep no track of the relationship between the features extracted by the convolutional layers, makes them less appealing and strongly criticised by front end researchers like Geoffrey Hinton. In order to avoid the limitations of pooling operations, it is suggested to use dynamic routing (routing-by-agreement) scheme, in replacement of the max-pooling operation and name this newly proposed model as Capsule Network (Sabour, Frosst & Hinton, 2017). Springenberg et al. (2014) also proposed to discard the pooling layer in favour of architecture that only consists of repeated convolutional layers. In order to reduce the size of the representation, he suggested using larger stride in convolutional layer once in a while. Discarding pooling layers has also been found important in training good generative models, such as variational autoencoders (VAEs) or generative adversarial networks (GANs) (Yu et al., 2017). From these moves, it seems likely that the future architectures will feature very few to no pooling layers.

Keeping in view these recent trends of research to improve deep models as classifiers, we hereby take inspiration from the characteristics of local binary pattern (LBP) operator, known widely for its simplicity and discriminative power to improve the representational power of CNN’s intermediate layers and utilise it for gaining better discrimination performance on image classification task. Similar work has been carried out by Juefei, Boddeti & Savvides (2017), who proposed an efficient non-linear approximation of convolutional layers in the convolutional neural network. Their proposed model namely local binary convolutional neural networks (LBCNN) (Juefei, Boddeti & Savvides, 2017) utilises a hybrid combination of fixed sparse and learnable weights and local binary patterns (LBP). In contrast, this work deploys dense weights and resides on regularisation techniques like dropout and batch normalisation to avoid overfitting issues.

Preliminaries

Local binary patterns

Local binary pattern (LBP) is a non-parametric approach that extracts local features of images by comparing the intensity of each center pixel in a patch with adjacent pixels in its defined neighbourhood (Ojala, Pietikainen & Harwood, 1994). If the neighbours have intensity greater than the center pixel, they are assigned the value of 1, otherwise 0. LBP has traditionally worked well with window patches of size 3 × 3, 5 × 5 and 7 × 7, etc, scanned through the image in an overlapping fashion. This bit string is read sequentially in a specified order and is mapped to a decimal number (using base 2) as the feature value assigned to the central pixel. These aggregate feature values represent the local texture in the image. The parameters and configurations of LBP could be tweaked by customising the window size, base, pivot (pixel treated as physical center of the patch) and ordering (clockwise/anticlockwise encoding).

Convolutional neural networks (CNN)

Convolutional neural network (CNN) is a multi-layered feed forward artificial neural network consisting of neurons in different layers to detect high level features from visual patterns automatically. Unlike the traditional feature extraction approaches where the features are hand engineered, CNN draws the features automatically by retaining their temporal and spatial information. The classical architecture of CNN consists of the following layers: (a) Input layer, (b) Convolutional layer, (c) Pooling layer, (d) Fully Connected/Dense layer and (e) Output layer. Except for the input and output layers, the remaining layers change their order and count giving rise to various types of neural architectures.

Ever since the successful exhibit of CNN for large scale image classification and retrieval (Krizhevsky, Sutskever & Hinton, 2012), various architectures of CNN have been proposed that alter the hidden layers’ order, count, types of activation functions and learning algorithm to improve the model’s discrimination performance and retrieval speed. We have chosen two popular architectures: LeNet and AlexNet to showcase the efficacy of the proposed approach on benchmark data sets. LeNet is the pioneering neural network proposed by Yann LeCun consisting of seven layers (five hidden), and is known to work very well for recognising digits and zip codes (LeCun et al., 1998). AlexNet, named after Krizhevsky, Sutskever & Hinton (2012), is a groundbreaking CNN consisting of five convolutional and three fully connected layers showing outstanding performance on large scale image recognition data set. The two architectures are demonstrated in Fig. 1. The gradient of CNN’s cost function is computed through backpropagation algorithm and the model parameters are updated through stochastic gradient descent (SGD) learning algorithm.

Figure 1 Classical CNN Architectures: LeNet and AlexNet used to outperform the state of the art image classification results on MNIST and ImageNet data sets.

Methodology

In order to enhance the discrimination power and representation capability of intermediate layers in CNN, we reformulate its architecture by introducing a discriminatively boosted alternative to pooling (DBAP) layer embedded at early stage of feature learning. Figure 1 demonstrates how LeNet and AlexNet models stack convolutional and pooling layers to learn local spatial features. We first preprocess each input image by performing standardization approach. The goal of standardization is to bring all the features at the same scale so that each feature is treated equally important and none dominates the other during features learning. Each image pixel xi(j) is standardized by computing the mean, μi and standard deviation, σi of each feature i in an image j by utilising the following formula:

(1) xi(j)=xi(j)−μiσi

Standardizing input data is a common approach used in neural networks and machine learning in general, to learn parameters, optimise and converge the models faster (Xiang & Li, 2017). After doing standardization, the d dimensional features are passed to the convolutional layer to capture the local features of the image. This result is next passed to the activation function to map the learned features in a non-linear space. Conventionally, the CNN architecture forward propagates the result of activation functions to a pooling layer that uses 2 × 2 filter window to down sample the features detected in non-linear space. The proposed framework replaces the first pooling layer of CNN with an alternative layer named as discriminatively boosted alternative to pooling (DBAP) layer. See Fig. 2 for illustration of the proposed changes in the CNN architecture. The DBAP layer takes its inspiration from local binary pattern (LBP) operator that acts as a powerful descriptor to summarise the characteristics of local structures in an image. The layer processes the features received from the previous layer by following the steps outlined in Algorithm 1. A 3 × 3 window with replicated boundary pixel padding is deployed to capture the local features of the image. Each pixel in the image is treated as a pivot (center pixel) to replace its intensity in connection to the intensity of pixels in its surrounding defined by the filter window. For each image patch, the neighbouring pixel values acquire the value 1 if their magnitude is equivalent or greater than the magnitude of the centre pixel. The magnitude is taken as 0 otherwise. For the example demonstrated in Fig. 2, the resulting LBP value for the center pixel is 11000111, equivalent to 227 in decimal number system. We move the filter one stride forward to compute LBP feature for each pixel in the image. For the given filter size, the DBAP layer computes 8-bit binary values for all the image pixels and converts them into their decimal equivalent. These values are totally based on the properties of the pixels in relationship to their neighbours. Our proposed DBAP layer is non-parametric and extracts more discriminative and visually powerful features as compared to the maxpooling layer used in benchmark CNN architectures. After processing the data through DBAP layer, it is forward propagated to the next layers in each architecture (LeNet and AlexNet) and treated in a conventional manner. In LeNet, this information passes on to the following layers in sequence: Convolutional, Pooling, Fully Connected, Fully Connected, and Fully Connected layers, whereas in AlexNet, the flow of information after DBAP takes the following route in sequence: Convolutional, Pooling, Convolutional, Convolutional, Convolutional, Pooling, Fully Connected, Fully Connected, Fully Connected layers. We discuss the implementation details regarding CNN model’s training and testing in “Experiments and Results”.

Figure 2 Graphical abstract of DBAP layer embedded in classical convolutional neural network models for boosting discrimination performance and feature visualisation power.

Algorithm 1 Discriminatively boosted alternative to pooling (DBAP) layer in CNN.

Input: Input Image, X(j)={xi(j)}di =1; Filter Size, F; Stride, S; Number of Neighbours, P; Index of Neighbour, p; Kernel, K.
Output: DBAP features, 1 × d

1: while not converge do
2:  for Each Image do
3:   Mean normalise the incoming image pixels X(j) and store them in X(j)norm.
4:   Compute the convolutional features from normalised image X(j)norm by convolving kernel K.
5:   Apply activation function on convolved features to map them in non-linear space.
6:   Forward propagate the non-linear result of activation function to DBAP layer.
7:   Partition the received image into overlapping blocks of equal size using the stride, S and filter size, F.
8:   Compute the LBP for each block using formula:
9:   LBPR,P=∑p=0P−1s(gp−gc).2p,
where s(gp − gc) = 1 if gp ≥ gc, 0 otherwise.
% Here gp and gc denote the gray values of the central pixel and its neighbours.
10:   Concatenate all the feature blocks represented by DBAP layer and forward pass the learned features in vectorised form to the next layer in CNN.
11:   Continue forward pass and perform backpropagation to learn model parameters.
12:  end for
13: end while	

Experiments and results

Data sets used

We have evaluated the efficacy of the proposed approach on different benchmark data sets with baseline convolutional neural networks and their other very deep counterparts such as GoogleNet (Szegedy et al., 2015), LBCNN (Juefei, Boddeti & Savvides, 2017) and MobileNet (Howard et al., 2017). There are four standard data sets used in this paper: MNIST, SVHN, FASHION-MNIST and CIFAR-10. These are benchmark computer vision data sets that are well understood and highly used by the researchers to provide basis for any improvement in the proposed learning algorithm or neural architecture. Their popularity has won them a regular place in many deep learning frameworks such as Keras, TensorFlow and Torch. Consequently, their off the shelf use is constantly on the rise, more than PASCAL VOC and ImageNet data sets till date (https://trends.google.com/trends/explore?date=all&q=mnist,%2Fg%2F11gfhw_78y,SVHN,%2Fg%2F11hz37p042,Imagenet).

The Modified National Institute of Standards and Technology (MNIST) data set (LeCun et al., 1989) consists of 60,000 training and 10,000 test images of hand written digits with a resolution of 28 × 28 pixels. The database contains grayscale images of digits 0 to 9. Despite the success of deep models with large scale data sets, MNIST enjoys the title of most widely used test bed in deep learning, surpassing CIFAR 10 (Krizhevsky & Hinton, 2009) and ImageNet (Deng et al., 2009) in its popularity via Google trends (https://trends.google.com/trends/explore?date=all&q=mnist,CIFAR,ImageNet). We have therefore selected this data set to benchmark the results of our proposed approach with state of the art comparative methods.

The FASHION-MNIST (F-MNIST) data set (Xiao, Rasul & Vollgraf, 2017) comprises of 28 × 28 grayscale images of 70,000 fashion products belonging to 10 different categories: TShirt/Top, Trouser, Pullover, Dress, Coat, Sandals, Shirt, Sneaker, Bag, Ankle boot. Similar to MNIST, the training set of FASHION-MNIST also comprises of 60,000 train images and 10,000 test set images.

The Street View House Numbers (SVHN) (Netzer et al., 2011) is a real world image data set consisting of digits in natural scenes of street houses. The digits 0 to 9 offer a multi-class classification problem with spatial resolution of 32 × 32 pixels. The data distribution consists of 73,257 train digits and 26,032 test digits for performance evaluation. These images show vast intra-class variations and include complex photometric distortions making the recognition problem a challenge just as in a general-purpose object recognition or natural scene understanding system.

The CIFAR-10 data set (Krizhevsky, Nair & Hinton, 2014) contains 60,000 color images from 10 different classes: Trucks, cats, cars, horses, airplanes, ships, dogs, birds, deer and frogs. The images have spatial dimension of 32 × 32 pixels. The data set consists of five training batches with each batch comprising of 10,000 train images. The test batch contains 10,000 images with 1,000 randomly-selected images from each class.

Tools used and computational requirements of the proposed model

The proposed neural model with DBAP layer was trained on Google Colab’s (Google Colab, 2019) Tesla K-80 graphics processing unit (GPU) using Keras (Chollet, 2015) and TensorFlow deep learning frameworks implemented in Python. Colab is a cloud based service that allows researchers to develop deep learning applications with free GPU support. The system used had Intel(R) Xeon(R) 2.3GHz processor with two cores and 16GB of RAM. To achieve results in optimal time, it is recommended to run the deep learning framework on premium GPU cards with at least 8 GB of RAM.

Evaluation metrics used for monitoring classification performance

The evaluation metrics used to monitor the quality of classification framework are accuracy, precision, recall, F1-score, and, area under the curve (AUC). These are standard model evaluation metrics used in research to carry out investigation and perform analysis (Le, 2019; Do, Le & Le, 2020). Accuracy is not regarded as a good measure of judging model’s performance when the class distribution is imbalanced, i.e. when the number of samples between two or more classes vary significantly. Such imbalance can affect the traditional classifiers as well as the deep models, commonly resulting in poor performances over the minority classes. Since, class instances of all the data sets used in this work are not balanced (in specific SVHN), we have demonstrated precision, recall, F-1 score, and receiver operating characteristics (in addition to accuracy to judge the performance of the proposed features and classifiers).

Visual diagnostics used to evaluate feature information quality

In order to understand how the input image is transformed by each intermediate layer of CNN, the activations of neurons in pooling layer and DBAP layer are visualised. The feature maps are visualised in three dimensions: Width, height and depth (channels). Since each channel encodes independent information, one appropriate way to visualise these features is to plot 2D images of each channel separately. Given our existing knowledge of deep neural models, the initial layers act as edge detectors and retain most of the information of the input image. As we go higher, the activations become increasingly abstract and less interpretable visually. The sparsity of activations increases with the depth of the layer, i.e. more and more filters would go blank and the pattern encoded in the image could not be seen. We thus expect that the activation filters of DBAP layer should be more interpretable and semantically meaningful given the input image, model is observing.

Implementation details for model training

In this section, we discuss how the choice of different hyper-parameters such as kernel’s filter size, batch size, learning rate, epochs and optimisation algorithm is made to train the CNN models for each specific data set on board. To decide on this, we first divide our data set into three different subsets: Train set, cross validated set and test set. For the selected benchmark data sets discussed in “Data Sets Used”, the train and test set segregation exists already. The cross validated set is obtained by splitting the train data randomly in 80:20 ratio, reserving 20% of the data points for the validation purpose and 80% of the train instances for the training objective. When deciding optimal values of epochs, learning rate, batch size, filter size and optimiser, 80% of these train instances are used to train both the neural models and their performance is judged on the 20% validation set examples. Once optimal values of these parameters are decided, the entire train set is used to train both the neural models and their performance is assessed on the available test sets. The train time of the proposed CNN models varies within this wall clock range (2.5, 3 h), when run on Google Colab.

In order to assess if the model is overfitting with the chosen set of parameters or hyper-parameters, the performance is compared on train and validation sets in Figs. 3 and 4. If the model behaves very well on the train set but fails to classify examples from the validation set by a huge margin, it means that it is overfitting and shall not perform well on unseen test examples. Some of the ways in which model overfitting could be avoided are: cross-validation, usage of more train data, early stopping, regularisation and removal of features. We have regularised the models which were overfitting with the help of the validation set.

Figure 3 Train and test accuracy curves of LeNet with DBAP layer are demonstrated on state-of-the art benchmark data sets: (A) MNIST, (B) CIFAR-10, (C) FASHION-MNIST and (D) SVHN.

The softmax activation function is used to enable LeNet for classification task.

Figure 4 Train and test accuracy curves of AlexNet with DBAP layer are demonstrated on state-of the-art benchmark data sets: (A) MNIST, (B) CIFAR-10, (C) FASHION-MNIST and (D) SVHN.

The softmax activation function is used to enable AlexNet for classification task.

Impact of learning rate and epochs on model training

The training of CNN depends largely on the learning rate and number of epochs used to learn the parameters. The learning rate hyperparameter controls the speed at which the model learns. For small learning rate, large number of epochs are required to train the model, whereas for large learning rate, small number of epochs are needed to navigate in the parameter space of the neural model. A learning rate that is too large can cause the model to converge too quick to a sub-optimal solution, whereas a learning rate that is too small can cause the learning process to become very slow. Therefore, it is advised to choose a value that is neither too large nor too small. Its value typically ranges between 0 and 1. We have configured the best value for learning rate using grid search method. Grid search involves picking values approximately on a logarithmic scale within the set range: {10−4, 10−3, 10−2, 10−1, 100}, and observes the validation loss while keeping the value of epochs fixed. We confined the value of epochs to 50 and observed the impact of changing learning rate on the validation set. Figures 3 and 4 demonstrate the accuracy of LeNet and AlexNet models, when the learning rate was fixed at 0.01 and the model was run for 50 epochs. Since the validation error is lowest when η = 0.01, and the gap between the train and validation error is not significantly large, the model does not tend to overfit and 0.01 turns out to be the most suitable value for learning rate.

Impact of batch size on model training

Batch size is also an important hyperparameter that impacts a model’s performance. Table 1 shows the best batch size for each data set when learning rate and epochs are fixed at 0.01 and 50 respectively using the AlexNet architecture. A similar comparison was also performed for LeNet architecture and best batch sizes for MNIST, Fashion-MNIST, SVHN and CIFAR-10 were chosen as 128, 128, 128 and 256 respectively.

Table 1 Overall accuracy of the proposed system on the validation set using different batch sizes.

For each data set, the optimal batch size could be seen via the best accuracy shown in bold.

Data sets	Batch sizes	
	64 (%)	128 (%)	256 (%)	
MNIST	99.1	98.6	99.5	
FASHION-MNIST	90.8	90.2	91.8	
SVHN	94.4	92.5	93.1	
CIFAR-10	78.3	78.8	80.6	

Impact of optimisers

In order to update the parameters of convolutional neural network, different popular optimisers such as stochastic gradient decent (SGD), adam (Kingma & Ba, 2014) and ADADELTA (Zeiler, 2012), were tested and evaluated on the validated set. Table 2 highlights the accuracy of AlexNet with DBAP layer when different types of optimisers were used. We observe that for MNIST data set, ADADELTA optimiser shows the best results, whereas for FASHION-MNIST, SVHN and CIFAR-10 data sets, SGD optimiser outperforms the remaining optimisation algorithms. A similar analysis was also performed for LeNet with DBAP layer and best optimisers were selected accordingly.

Table 2 Overall accuracy of the proposed system on the validation set using different types of optimisers for training AlexNet.

For each data set, the optimal optimiser varies based on the best accuracy shown in bold.

Data sets	Optimisers	
	SGD (%)	Adam (%)	AdaDelta (%)	
MNIST	98.1	98.6	99.5	
FASHION-MNIST	91.5	90.6	90.1	
SVHN	93.2	92.1	91.7	
CIFAR-10	78.2	78.1	77.6	

Impact of LBP filter size on CNN

We have also assessed different kernel sizes used in DBAP layer to capture local features of images that add to the discriminative ability of neural models. Table 3 shows that 3 × 3 window gives best accuracy on the validation set in comparison to larger size filters on all the data sets.

Table 3 Classification accuracy on the validation set of four benchmark data sets with varying filter size in DBAP layer of AlexNet.

The results reveal that a 3 × 3 kernel is suitable than the rest.

Data sets	Window size	
	3 × 3 (%)	5 × 5 (%)	7 × 7 (%)	
MNIST	99.5	97.1	96.1	
F-MNIST	91.8	88.9	88.2	
SVHN	94.4	93.2	91.5	
CIFAR-10	80.6	74.5	76.2	

Model testing

After fine tuning the neural models with optimal parameters and hyperparameters, we next compute the classification performance of the proposed model on unseen test examples of each standard data set.

Analysis of CNN model with DBAP layer as a classifier

When deploying CNN as a classifier, the test data is passed to the trained CNN model with DBAP layer, whose last layer consisting of softmax units is utilised for object categorisation. The discrimination performance of the model is assessed with the help of following evaluation metrics: Accuracy, precision, recall, F1-score, and area under the curve (AUC), discussed in “Evaluation Metrics Used for Monitoring Classification Performance” and “Appendix”. Table 4 shows improvement in the discrimination performance yielded by the proposed approach in comparison to the baseline AlexNet and LeNet architectures on four different benchmark data sets. We have also compared our results with local binary convolutional neural network (LBCNN) that offers to provide an alternative to standard convolutional layers in the convolutional neural network (Juefei, Boddeti & Savvides, 2017), GoogleNet (also known as Inception V1) (Szegedy et al., 2015) and MobileNet (Howard et al., 2017). GoogleNet is a 22-layer CNN inspired by LeNet, whereas MobileNet is an efficient CNN architecture with 17 layers streamlined for mobile applications. We observe that the classification performance of the proposed model with DBAP layer is competitive to the state of the art results shown by ultra deep convolutional neural models. The precision, recall and F1 scores of the proposed model further reassure the precision and discrimination power of the proposed deep model for unseen test examples.

Table 4 Classification accuracy yielded by LeNet and AlexNet (in %) after incorporation of the DBAP layer is shown in bold.

The classifier used is softmax by both the models. One can observe that the results are better than those achieved by the baseline models and competitive to the discrimination results of other popular deep models.

Data Sets	Baseline LeNet (%)	LeNet with DBAP (%)	Baseline AlexNet (%)	AlexNet with DBAP (%)	LBP features with k-NN (%)	LBP features with SVM (%)	MobileNet (Howard et al., 2017) (%)	GoogLeNet (Szegedy et al., 2015) (%)	LBCNN (Juefei, Boddeti & Savvides, 2017) (%)	
MNIST	99.0	99.1	99.2	99.5	88.7	83.7	94.59	97.98	99.51	
F-MNIST	89.8	91.0	90.5	91.5	78.3	73.5	–	93.5	–	
SVHN	86.7	88.3	87.3	94.4	29.6	25.9	90.8	92.3	94.50	
CIFAR-10	72.3	74.8	73.7	80.6	28.3	27.6	65.6	76.5	92.99	

In Table 4, one may observe that unlike other data sets, the classification results of DBAP features on CIFAR-10 data set are a lot worse in comparison to LBCNN (Juefei, Boddeti & Savvides, 2017). This is because the images in CIFAR-10 possess natural objects with rich textures as compared to the hand written digit images present in other data sets. For this reason, LBCNN works exceptionally better on CIFAR-10 in comparison to AlexNet with DBAP features. Also LBCNN replaces all convolutional layers of AlexNet with LBP inspired layers which is popular for extracting discriminative texture descriptors, whereas our proposed model only replaces the first MaxPooling layer with LBP inspired feature detectors, hence the performance gap is higher in contrast. Similar impact in performance could also be observed in area under the curve graphs shown in the Appendix section.

We have conducted experiments to compare the discrimination power of LBP operator with DBAP features in Table 4. The classifiers used for the purpose are k-NN and SVM. One can observe that LBP operator on its own does not yield as good classification results as the DBAP layer introduced in LeNet and AlexNet architectures. The open source code developed for these experiments is available at https://github.com/shakeel0232/DBAP-CNN.

Analysis of CNN model with DBAP layer as a feature extractor

In order to assess the discrimination power of features learned by DBAP layer, we have also checked their accuracy with simple off the shelf classifiers like k-nearest neighbour (k-NN) and support vector machines (SVM). We selected pre-trained CNN models with and without DBAP layer to extract features for image classification task. The results shown in Tables 5, 6, 7 and 8 demonstrate that DBAP layer can serve as a competitive feature extractor in comparison to the intermediate layer features such as MaxPooling layer of AlexNet and LeNet. For SVM classifier, the optimal value of parameter C is searched via grid-search method on the validation set and shown against each data set in the tables. Similarly, for k-nearest neighbour (k-NN), the optimal value of k is searched using the validation set and then used for the test data in each benchmark data set. The empirical results reveal that DBAP features could be used as readily available features from a pre-trained model for applications where quick retrieval and classification results are required.

Table 5 Accuracy of SVM classifier on DBAP features derived from pre-trained LeNet with DBAP layer. The DBAP features show better classification results than the MaxPool features in LeNet. The fully connected (FC) layers of LeNet with DBAP also tend to show better discrimination ability as compared to FC layer features extracted from regular LeNet on all benchmark data sets.

Data sets	MaxPool layer (Layer 2)	DBAP layer (Layer 2)	FC layer from LeNet (Layer 7)	FC layer from LeNet with DBAP (Layer 7)	
MNIST	98.1% (C = 100)	98.3% (C = 100)	98.4% (C = 100)	99.0% (C = 1)	
F-MNIST	88.6% (C = 100)	89.0% (C = 100)	90.4% (C = 100)	91.3% (C = 100)	
SVHN	81.2% (C = 10)	82.0% (C = 10)	83.9% (C = 10)	86.8% (C = 100)	
CIFAR-10	52.1% (C = 10)	52.9% (C = 100)	57.4% (C = 10)	65.3% (C = 10)	

Table 6 Accuracy of SVM classifier on DBAP features derived from pre-trained AlexNet with DBAP layer. The classification results are better than the results obtained by MaxPool features derived from a regular AlexNet. The inclusion of DBAP layer also shows better FC features from the model giving better classification results in comparison to the FC features from regular AlexNet model.

Data sets	MaxPool layer (Layer 2)	DBAP layer (Layer 2)	FC Layer from AlexNet (Layer 11)	FC Layer from AlexNet with DBAP (Layer 11)	
MNIST	95.1% (C = 100)	98.0% (C = 100)	99.1% (C = 100)	99.2% (C = 0.01)	
F-MNIST	89.8% (C = 1)	90.4% (C = 1)	90.9% (C = 100)	91.4% (C = 1)	
SVHN	80.1% (C = 1)	80.2% (C = 100)	89.0% (C = 1)	94.5% (C = 10)	
CIFAR-10	60.3% (C = 100)	63.3% (C = 100)	80.3% (C = 100)	84.3% (C = 100)	

Table 7 Accuracy of k-NN classifier on DBAP features derived from pre-trained LeNet with DBAP layer. The results achieved are better than the ones obtained by MaxPool layer in a regular LeNet. The inclusion of DBAP layer also improves the FC features for discrimination task.

Data sets	MaxPool layer (Layer 2)	DBAP layer (Layer 2)	FC layer from LeNet (Layer 7)	FC layer from LeNet with DBAP (Layer 7)	
MNIST	97.4% (k = 2)	97.7% (k = 2)	98.8% (k = 2)	99.0% (k = 2)	
F-MNIST	77.6% (k = 2)	78.2% (k = 2)	84.6% (k = 2)	91.2% (k = 16)	
SVHN	77.2% (k = 32)	78.6% (k = 32)	85.9% (k = 2)	86.6% (k = 8)	
CIFAR-10	56.0% (k = 2)	59.7% (k = 9)	63.8% (k = 4)	65.0% (k = 27)	

Table 8 Accuracy of k-NN classifier on DBAP features derived from pre-trained AlexNet with DBAP layer. The classification results achieved are better than those obtained by MaxPool features derived from regular AlexNet. The inclusion of DBAP layer also improves the discrimination quality of FC features in AlexNet with DBAP layer.

Data Sets	MaxPool layer (Layer 2)	DBAP layer (Layer 2)	FC Layer from AlexNet (Layer 11)	FC Layer from AlexNet with DBAP (Layer 11)	
MNIST	97.9% (k = 2)	98.0% (k = 2)	98.7% (k = 2)	99.1%(k = 4)	
F-MNIST	83.1% (k = 2)	87.2% (k = 2)	88.6% (k = 2)	89.5% (k = 16)	
SVHN	66.0% (k = 2)	68.4% (k = 2)	88.0% (k = 2)	94.4% (k = 16)	
CIFAR-10	52.7% (k = 32)	53.6% (k = 41)	68.9% (k = 16)	83.7% (k = 16)	

We have also assessed the impact of DBAP layer on FC layer features. The fully connected (FC) layers are known to retain better discrimination power for classification tasks, however with the inclusion of DBAP layer, their ability to classify objects is further improved as can be seen in the last two columns of Tables 5, 6, 7 and 8.

Statistical significance of models

We have also applied hypothesis testing to estimate the statistical significance of the proposed models. Statistical tests help us identify the behaviour of models if the test set changes. Since our data sets are standardised, we assume a normal distribution of features and have applied McNemar’s test or 5 × 2 cross-validation with a modified paired Student t-test. The null hypothesis assumes that the two samples came from the same distribution. In contrast, the alternative hypothesis assumes that the samples came from two different distributions and hence there is a difference between the tested models or classifiers. With 0.05 level of confidence/significance, the p values attained for LeNet with DBAP layer and AlexNet with DBAP layer models are 0.007 and 0.011 respectively. In both the cases, p < 0.05, shows the samples generated from the proposed architectures are statistically different from the ones without DBAP layer.

Visualisation of filters

We have also visualised the mid-level features learned by DBAP layer and compared them with the features learned by max-pooling layers used in classical CNN architectures. Figures 5 and 6 demonstrate the improvement in visual representation of intermediate features learned by the two CNN architectures in comparison to their baseline counterparts with maxpooling layer. One can observe that DBAP layer learns semantically better features from the input images as compared to the maxpooling layer used in classical LeNet and AlexNet architectures. As we go higher in the model hierarchy, the filters become more abstract and sparsity of the activations increases, i.e. the filters become more blank and the pattern encoded by the image is not showcased by the filter (François, 2017).

Figure 5 Visualising the response of neurons in the MaxPool layer and DBAP layer present in baseline LeNet and LeNet with DBAP layer respectively.

With six filters/kernels deployed in the first MaxPool layer of LeNet, one can observe that the visualisations of DBAP layer demonstrate more meaningful information about the input image as compared to the MaxPool layer. The input images used here are: (A) Digit three from MNIST data set, (D) Shirt from FASHION-MNIST data set, (G) Digit forty nine from SVHN data set, (J) Dog’s image from CIFAR-10 data set. (B), (E), (H) and (K) show neuron activations of the first maxpool layer in LeNet and (C), (F), (I) and (L) show neuron activations of DBAP layer in LeNet.

Figure 6 Visualising the response of neurons in MaxPool layer and DBAP layer with baseline AlexNet and AlexNet with DBAP layer respectively.

AlexNet uses 96 filters/kernels of size 3 × 3 in the first MaxPool layer and one can see that DBAP layer retains most of the input image’s information as compared to the MaxPool layer. The input images used here are: (A) Digit three from MNIST data set, (D) Shirt from FASHION-MNIST data set, (G) Digit forty nine from SVHN data set, (J) Dog’s image from CIFAR-10 data set. (B), (E), (H) and (K) show neuron activations of the first maxpool layer in AlexNet and (C), (F), (I) and (L) show neuron activations of DBAP layer in AlexNet.

Improving the visualisation strength of neural models can help us explore and understand the black box learning behaviour of deep models. Better visualisation can serve as a great diagnostic tool (Liu, Zeng & Gifford, 2019) for observing the evolution of features during model training and diagnose potential problems with the model via online/offline feature representations. This facilitates the researchers to fix their training practices and find models that can outperform an existing successful deep model. For example, the deconvolutional technique proposed for visualising the hidden layer features suggested an architectural change of smaller convolutional filters that lead to state of the art performance on the ImageNet benchmark in 2013 (Zeiler & Fergus, 2014).

Proposed model’s complexity

We next compare the count of trainable parameters in LeNet and AlexNet containing DBAP layers with their baseline counter parts in Table 9. The total number of CNN parameters are the sum of all its weights and biases connecting the convolutional, input, output and fully connected layers. The pooling layers in the architecture do not contribute to the count of model parameters as they contain hyper-parameters such as pool size, stride, and padding which do not need to be learned during the training phase. The number of model parameters before the advent of DBAP layer remain fixed. However, when we replace the first pooling layer with DBAP layer, the output tensor of Layer 2 is not down sampled as it does in regular LeNet and AlexNet architectures, rather the tensor scale remains the same as its input (i.e. 26 × 26 × 6 for LeNet and 14 × 14 × 96 for AlexNet). This impacts the size of the kernel in the following convolutional layer, and the effect is carried out forward to the next maxpooling and fully connected layers. Overall, there is an increase of 380.33% in LeNet parameters and an increase of 14.57% in AlexNet model parameters with the inclusion of DBAP layer.

Table 9 Comparison of the number of trainable parameters in regular LeNet, LeNet with DBAP layer, regular AlexNet and AlexNet with DBAP layer.

Layer name	Tensor size	Number of parameters	
Input image	28 × 28 × 1	0	
Conv-1	26 × 26 × 6	60	
MaxPool-1	13 × 13 × 6	0	
Conv-2	11 × 11 × 16	880	
MaxPool-2	5 × 5 × 16	0	
FC-1	120 × 1	48,120	
FC-2	84 × 1	10,164	
FC-3	10 × 1	850	
Output	10 × 1	0	
Total		60,074 (∼0.06M )	
(b) Architecture of LeNet with DBAP layer	
Input image	28 × 28 × 1	0	
Conv-1	26 × 26 × 6	60	
DBAP	26 × 26 × 6	0	
Conv-2	24 × 24 × 16	880	
MaxPool-2	12 × 12 × 16	0	
FC-1	120 × 1	276,600	
FC-2	84 × 1	10,164	
FC-3	10 × 1	850	
Output	10 × 1	0	
Total		288,554 (∼0.28M )	
(c) AlexNet architecture	
Input image	28 × 28 × 1	0	
Conv-1	14 × 14 × 96	960	
MaxPool-1	7 × 7 × 96	0	
Conv-2	7 × 7 × 256	614,656	
MaxPool-2	3 × 3 × 256	0	
Conv-3	3 × 3 × 384	885,120	
Conv-4	3 × 3 × 384	1,327,488	
Conv-5	3 × 3 × 256	884,992	
MaxPool-3	1 × 1 × 256	0	
FC-1	4,096 × 1	1,052,672	
FC-2	4,096 × 1	16,781,312	
FC-3	10 × 1	40,970	
Output	10 × 1	0	
Total		21,588,170 (∼21M )	
(d) Architecture of AlexNet with DBAP layer	
Input image	28 × 28 × 1	0	
Conv-1	14 × 14 × 96	960	
DBAP	14 × 14 × 96	0	
Conv-2	14 × 14 × 256	614,656	
MaxPool-2	6 × 6 × 256	0	
Conv-3	6 × 6 × 384	885,120	
Conv-4	6 × 6 × 384	1,327,488	
Conv-5	6 × 6 × 256	884,992	
MaxPool-3	2 × 2 × 256	0	
FC-1	4,096 × 1	41,98,400	
FC-2	4,096 × 1	16,781,312	
FC-3	10 × 1	40,970	
Output	10 × 1	0	
Total		24,733,898 (∼24M )	

Keeping in view the size of model parameters, the proposed model is not well suited for resource constrained environments, where storage and computation of large number of parameters becomes a bottleneck. However, it offers two fold advantage in comparison to the state of the art models: (1) Effective intermediate feature visualisation power and (2) competitive discrimination performance as a feature extractor and classifier. Models such as LBCNN (Juefei, Boddeti & Savvides, 2017) propose to use a compact neural model whose convolutional layers are all replaced by LBP operator. This move reduces the number of learnable parameters massively to around 0.352 million, thus making it very suitable for resource constrained environments.

Conclusion & future work

In this paper, we propose to induce discrimination into the intermediate layers of the convolutional neural network by introducing a novel local binary pattern layer that can serve as a replacement of the first standard maxpooling layer used at early stage of feature learning in the convolutional neural network. The empirical results on benchmark data sets as well as the visual feature maps of intermediate layers demonstrate the strength of the proposed idea to learn more discriminative features without building ultra deep models. Our experiments reveal that the proposed approach can strengthen the discriminative power of mid-level features as well as high level features learned by fully connected (FC) layers of convolutional neural network. The experiments with simple classifier like k-NN and popular industry classifier like SVM, suggest the use of intermediate DBAP layer and its following fully connected layers in the deep learning pipeline for off-line feature extraction and classification tasks.

In future, we aim to improve the training complexity of the proposed approach by reducing the number of learnable parameters for model training. In this regard, we shall explore sparsity in the neural connections to adopt suitable regularisation technique for fast model learning.

Appendix

The Appendix section shows some additional results to support reproducible research and make the main text more readable and understandable. We have shown precision, recall and F1-score of LeNet and AlexNet models along with their improved counterparts in Tables 10 and 11. These evaluation metrics in combination with the accuracy show how good the proposed models are in comparison to their baseline models.

Table 10 Precision, recall and F1-score of LeNet and LeNet with DBAP layer using softmax classifier.

Data sets	LeNet	LeNet with DBAP	
	Precision (%)	Recall (%)	F1-score (%)	Precision (%)	Recall (%)	F1-score (%)	
MNIST	99	99	99	99	99	99	
FASHION-MNIST	90	90	90	91	91	91	
SVHN	86	85	86	88	87	87	
CIFAR-10	72	72	72	74	75	74	

Table 11 Precision, recall and F1-score of AlexNet and AlexNet with DBAP layer using softmax classifier.

Data sets	AlexNet	AlexNet with DBAP	
	Precision (%)	Recall (%)	F1-score (%)	Precision (%)	Recall (%)	F1-score (%)	
MNIST	99	99	99	100	99	100	
FASHION-MNIST	91	91	91	91	91	91	
SVHN	90	90	90	94	94	94	
CIFAR-10	71	72	71	77	78	77	

One can also observe the area under the curve (AUC) for the developed classifiers in Figs. 7, 8, 9 and 10. AUC ranges between 0 and 1. Higher the AUC, better the model is at predicting classes correctly as positive and negative, significantly above the random chance. AUC is good at catching the performance of models when the class distribution is skewed. We observe that with the addition of DBAP layer in CNN architecture, AUC in ROC either increases or remains the same as shown in few cases.

Figure 7 ROC Curve of k-NN, Softmax and SVM Classifiers on MNIST Data Set.

(A) AUC with Baseline AlexNet = 0.97 & AlexNet with DBAP layer = 0.98, (B) AUC with Baseline AlexNet and AlexNet with DBAP layer = 0.96, (C) AUC with Baseline AlexNet and AlexNet with DBAP layer = 0.97.

Figure 8 ROC Curve of k-NN, Softmax and SVM Classifiers on MNIST Data Set.

(A) AUC with Baseline AlexNet = 0.97 & AlexNet with DBAP layer = 0.98, (B) AUC with Baseline AlexNet and AlexNet with DBAP layer = 0.96, (C) AUC with Baseline AlexNet and AlexNet with DBAP layer = 0.97.

Figure 9 ROC Curve of k-NN, Softmax and SVM Classifiers on SVHN Data Set.

(A) AUC with Baseline AlexNet = 0.63 & AlexNet with DBAP layer = 0.66, (B) AUC with Baseline AlexNet = 0.57 and AlexNet with DBAP layer = 0.58, (C) AUC with Baseline AlexNet = 0.54 and AlexNet with DBAP layer = 0.55.

Figure 10 ROC Curve of k-NN, Softmax and SVM Classifiers on CIFAR-10 Data Set.

(A) AUC with Baseline AlexNet = 0.64 & AlexNet with DBAP layer = 0.65, (B) AUC with Baseline AlexNet = 0.66 and AlexNet with DBAP layer = 0.68, (C) AUC with Baseline AlexNet and AlexNet with DBAP layer = 0.68.

Additional Information and Declarations

Competing Interests

Author Contributions

Data Availability

The authors declare that they have no competing interests.

Shakeel Shafiq conceived and designed the experiments, performed the experiments, analyzed the data, performed the computation work, prepared figures and/or tables, and approved the final draft.

Tayyaba Azim conceived and designed the experiments, analyzed the data, authored or reviewed drafts of the paper, and approved the final draft.

The following information was supplied regarding data availability:

The Python code that trains and analyses the proposed model is available at GitHub: https://github.com/shakeel0232/DBAP-CNN

The MNIST Data Set is available at: http://yann.lecun.com/exdb/mnist/.

The Fashion MNIST is available at: https://github.com/zalandoresearch/fashion-mnist.

The SVHN Data Set is available at: http://ufldl.stanford.edu/housenumbers/.

The CIFAR-10 Data Set is available at:

https://www.cs.toronto.edu/~kriz/cifar.html.

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
