# Peer review of "Introspective analysis of convolutional neural networks for improving discrimination performance and feature visualisation"

_PeerJ Computer Science, doi:10.7717/peerj-cs.497_

## Round 0.1 · original submission · Major Revisions

Dear Authors,

Please carefully revise the manuscript according to the reviewer comments, please also improve the language and structure of the paper.

Thanks

Reviewer 1 ·

Basic reporting

- The graphical representation needs to be improved in Figure 1. The arrow sizes are different, and some are not straight. Does arrow length carry any information? If not, they should all be of the same size. Traditionally, the neural network layers are depicted from left to right, such as the schematic shown in Figure 2. Therefore, it is recommended to demonstrate the two CNN architectures in Figure 1 from left to right.
- The English language should be improved in Lines 117-132 to ensure your points are clearly described in a scientific language.
- Line 177: The terms normalization and standardization are sometimes used interchangeably, but they usually refer to different things. Normalization usually means to scale a variable to have a value between 0 and 1, while standardization transforms data to have a mean of zero and a standard deviation of 1 (z-score). Hence, it is recommended to use “standardization” instead of “normalization”.
- I recommend reporting some of the tables presented in the main manuscript to supplementary data so that the key points are highlighted, and additional results are available for an interested reader.

Experimental design

no comment

Validity of the findings

- One key missing analysis on the performance evaluation is the significance test in comparison with previous architectures such as LeNet and AlexNet in Table 5-10. Reporting p-value will provide additional information about the effectiveness and significance of the proposed approach.

Additional comments

- This is a very well written manuscript describing a novel approach for improving discrimination performance and feature visualization in Convolutional Neural Networks (CNN). Different parameters are taken into account and the results are reported with sufficient details and discussions.

Reviewer 2 ·

Basic reporting

- English language should be improved.

- It is necessary to provide more literature review related to the datasets that they used.

- Deep learning (i.e., CNN) has been used in previous works in a variety of fields such as https://doi.org/10.7717/peerj-cs.177 and https://doi.org/10.1016/j.neucom.2019.09.070. Therefore, it is suggested to provide more references in this description.

Experimental design

- How did the authors perform hyperparameter tuning in the models? It should be described clearly. Did the cross-validation technique apply?

- Hardware specification to train model should be described.

- The authors should release source codes for replicating the results.

- There is a need to refer to some works related to measurement metrics (i.e., recall, precision, accuracy, ...) of deep learning models such as PMID: 31362508 and PMID: 32613242.

- In addition to the aforementioned metrics, the authors should report ROC curves and AUC values for prediction models.

Validity of the findings

- The authors should have some validation data on the models.

- It looks like some models contained a lot of overfitting after few epochs (i.e., Fig. 3-4). How did the authors address this critical point?

- Training time should be reported.

- The authors should perform some statistical tests to compare the performance results among different models.

- The authors should compare the predictive performance with previous works that focused on the same datasets.

Additional comments

No comment

---

## Round 0.2 · accepted · Accept

Thanks for the revised submission, good luck

Reviewer 2 ·

Basic reporting

No comment.

Experimental design

No comment.

Validity of the findings

No comment.

Additional comments

My previous comments have been addressed well.